# Efficacy of Telemedicine and At-Home Telemonitoring following Hospital Discharge in Patients with COVID-19

**DOI:** 10.3390/jpm12040609

**Published:** 2022-04-10

**Authors:** Roi Suárez-Gil, Emilio Casariego-Vales, Rosa Blanco-López, Fernando Santos-Guerra, Cristina Pedrosa-Fraga, Álvaro Fernández-Rial, Iria Íñiguez-Vázquez, María Mar Abad-García, Mercedes Bal-Alvaredo

**Affiliations:** 1Internal Medicine Department, Lucus Augusti University Hospital, 27003 Lugo, Spain; emilio.casariego.vales@sergas.es (E.C.-V.); cristina.pedrosa.fraga@sergas.es (C.P.-F.); jose.alvaro.fernandez.rial@sergas.es (Á.F.-R.); iria.iniguez.vazquez@sergas.es (I.Í.-V.); mercedes.bal.alvaredo@sergas.es (M.B.-A.); 2Day Hospital Nursing, Lucus Augusti University Hospital, 27003 Lugo, Spain; rosa.maria.blanco.lopez@sergas.es (R.B.-L.); maria.mar.abad.garcia@sergas.es (M.M.A.-G.); 3Information Systems Project Management Department, Galician Health Service, Regional Ministry of Health, 15703 Santiago de Compostela, Spain; fernando.santos.guerra@sergas.es

**Keywords:** COVID-19, telemedicine, hospital discharge

## Abstract

Aim: This work aims to evaluate the safety and utility of an at-home telemedicine with telemonitoring program for discharged COVID-19 patients. Methods: This is a retrospective cohort study of all patients discharged home in Galicia between 6 March 2020 and 15 February 2021. We evaluated a structured, proactive monitoring program conducted by the ASLAM (Área Sanitaria de Lugo, A Mariña y Monforte de Lemos) Healthcare Area team compared to patients discharged in the rest of the Autonomous Community of Galicia. Results: During the study period, 10,517 patients were hospitalized for COVID-19 and 8601 (81.8%) were discharged. Of them, 738 (8.6%) were discharged in ASLAM and 7863 (91.4%) were discharged in the rest of Galicia. Of those discharged in ASLAM, 475 (64.4%) patients were monitored. Compared to patients in the rest of Galicia, the group monitored via telemedicine had a significantly shorter first hospital stay (*p* < 0.0001), a lower readmission rate (*p* = 0.05), and a shorter second hospital stay (*p* = 0.04), with no differences in emergency department visits or 90-day all-cause mortality. Conclusion: A structured, proactive telemedicine with telemonitoring program for discharged COVID-19 patients is a safe, useful tool that reduces the mean length of hospital stay and readmissions.

## 1. Introduction

During the coronavirus disease 2019 (COVID-19) pandemic, telemedicine has proven to be a suitable tool for monitoring outpatients with multiple diseases [1]. In the context of patients with COVID-19—A disease which is still not well known that requires physical distancing and has entailed a significant overload of healthcare services—Telemedicine has been used in two different scenarios. First, it is used for the at-home monitoring of recently diagnosed patients. In this scenario, it has been observed that at-home monitoring is able to predict which patients are at high risk of requiring hospitalization [2], is efficacious and safe for the at-home management of many patients [3], and reduces patient visits to healthcare centers [4]. Second, it has been used for monitoring outpatients following discharge for patients hospitalized for COVID-19. This scenario is of particular interest as COVID-19 is a novel disease with a medium- and long-term prognosis that is not well known. However, analyzing experiences with this type of monitoring is not straightforward, given that there are not many programs and their approaches are highly heterogeneous. These programs may include not only patients in general hospitalization [5], but also patients from the emergency department or other hospital units [6]. Furthermore, monitoring may be done in the patient’s home, in medicalized hotels, or in nursing homes [7]. These factors make it difficult to establish the utility of telemedicine in the monitoring of this type of patients.

At the beginning of the pandemic, in March 2020, our center established a monitoring system for all patients following hospitalization due to COVID-19. It had two well-differentiated programs: one program was used when a patient was discharged home and another when a patient was discharged to a nursing home. This study aims to evaluate the clinical utility and safety of a standardized at-home monitoring program for patients after discharge from hospitalization due to COVID-19.

## 2. Materials and Methods

### 2.1. Study Design and Setting

This work is a retrospective cohort study of all patients with a positive PCR test for severe acute respiratory syndrome coronavirus 2 (SARS-CoV-2) who were discharged home in the Autonomous Community of Galicia between 6 March 2020 and 15 February 2021. Our healthcare network provides coverage for nearly all of its 2,701,819 residents and all its epidemiological, microbiological, and clinical information is centralized. In regard to the care model, this network is organized into seven healthcare areas, each of which encompasses hospitals (14 in total) and clinics (at least one in all of its towns). Among them, the Lugo, A Mariña, and Monforte de Lemos Healthcare Area (ASLAM, for its initials in Spanish) provides healthcare coverage to 345,000 residents and has three hospitals and 84 health centers.

### 2.2. Monitoring via Telemedicine

The steps and protocols used in the monitoring program for COVID-19-positive patients in the ASLAM (Área Sanitaria de Lugo, A Mariña y Monforte de Lemos) Healthcare Area have previously been reported [3,8]. In brief, at the start of the pandemic, a team comprising nurses and physicians with experience in telemedicine and telemonitoring programs was formed. One set of inclusion criteria for outpatients and another set of inclusion criteria for discharged patients were established. All patients monitored via telemedicine had daily monitoring whose specific characteristics depended on a risk stratification conducted by the medical team upon the patient’s inclusion in the program. The telemedicine tool used was “TELEA”, a platform the Galician Health Service (SERGAS, for its initials in Spanish) had previously used in programs for patients with chronic diseases that was adapted for this new situation. The TELEA platform adapted for patients with COVID-19 was used heterogeneously in the remaining six healthcare areas in Galicia outside of the ASLAM Healthcare Area, where it was used in a consistent, standardized way as detailed in this work.

### 2.3. Inclusion and Exclusion Criteria

In the ASLAM Healthcare Area, all patients discharged following hospitalization due to COVID-19 were eligible for monitoring. The following exclusion criteria were used to decide who would be monitored:Patients who had a very long hospital stay who were considered to have overcome the virus.SARS-CoV-2-positive patients without symptoms attributable to the virus who were hospitalized for another reason (such as a surgical intervention, for example).Patients whose attending physician considered that monitoring would provide no benefit given their clinical stability.Patients who refused to be included.Patients discharged to an institution, community health center, or who were monitored as part of home hospitalization.

The inclusion procedure was started when the attending physician reported each patient’s hospital discharge to the nursing department staff of a “virtual monitoring ward” created for the TELEA program. At that time, the relevant clinical data of each patient were collected. Upon a patient’s transfer home, he or she had telemonitoring equipment (pulse oximeter and thermometer) and instructions for accessing the virtual platform. In the initial hours following discharge, personnel from the virtual monitoring ward contacted the patient at home. At that time, they were able to verify one additional exclusion criterion: Technical issues that impeded participation in the program (for example, lack of internet at home).

The nursing department received and reviewed vital signs data (temperature and oxygen saturation) as well as the responses to a respiratory symptoms questionnaire every eight hours. In the event of incidents such as a change in biometrics or in the patient’s clinical condition as evaluated by the questionnaire [3,8], the nursing department team contacted the patient. If it was not possible to resolve the problem via telephone, a physician evaluated the situation and decided if it was necessary to transfer the patient to the hospital emergency department. In that case, the physician contacted the emergency department and explained the reasons underlying the referral and planned how to maintain at-home monitoring if hospital admission was not deemed necessary. In addition, the patient had a telephone number for direct contact with the nursing department team that was operative from 8:00 a.m. to 9:30 p.m. This monitoring protocol used for discharged patients is comparable to the closest type of monitoring used for outpatients [3].

### 2.4. Discharge from the TELEA Program

Monitoring via telemedicine following hospitalization was conducted for a minimum of ten days. Discharge from the program was always evaluated by a physician who contacted the patient via telephone on the tenth day of monitoring. The telemonitoring period could be extended if the following criteria for discharge from the program were not met:At least ten days have passed since the onset of symptoms.The patient does not have any symptoms or symptoms are residual.The patient has been afebrile for at least the last 72 hThe patient does not present with any other medical problems or complications.

In cases in which residual symptoms warranted epidemiological discharge from the preventative medicine department, monitoring via telemedicine was ended and the patient was given a priority appointment at a specialized “post-COVID-19” unit. To determine whether the monitoring of a specific case was appropriate, quality criteria were established that allowed for evaluating the degree of adherence (Table 1).

### 2.5. Ethical Aspects

The data were included in a registry approved by the ASLAM Healthcare Area Research Ethics Committee. During the initial interview, the nursing department staff of the virtual telemedicine ward explained the monitoring conditions, actions to be taken, possible risks, and how to gather precise data to the patients. Then, oral consent for inclusion in the program was requested from each patient. The sources of information included the patients’ monitoring data and data on their progress obtained from the centralized electronic medical records of the Galician Health Service (SERGAS).

### 2.6. Statistical Analysis

The usual descriptive statistics techniques were used for analyzing the data. The chi-square test was used to compare qualitative variables and Fisher’s exact test was used when required. Student’s *t*-test was used to compare two means after evaluating homoscedasticity. The level of statistical significance was established as *p* < 0.05. Analyses were conducted using the SPSS v18 statistics program (SPSS Inc., Chicago, IL, USA).

## 3. Results

A total of 105,257 patients were diagnosed with COVID-19 in Galicia between 6 March 2020 and 15 February 2021. Of them, 10,517 (9.9%) required hospital admission, of which 8601 (81.8%) were discharged, 1524 (14.5%) died during their hospital stay, and 392 (3.7%) remained hospitalized or in other circumstance at the time of analysis. Regarding distribution of those discharged according to healthcare area, 738 (8.6%) were discharged in ASLAM and the remaining 7863 (91.4%) were discharged in the other six healthcare areas in Galicia.

Of the 738 patients discharged in ASLAM, 551 (74.7%) met the inclusion criteria and should have been monitored via TELEA (Figure 1). However, only 475 patients were included in the TELEA program; the remaining 76 patients were not included for undetermined reasons. In regard to the other 187 cases that did not meet the inclusion criteria, 54 (7.4%) were considered not to require monitoring following a long hospital stay and cure; 123 (16.7%) were monitored from nursing homes, other hospitals, or at home by the home hospitalization department; and nine did not meet the inclusion criteria for other reasons. Table 2 summarizes these patients’ main characteristics.

In regard to the characteristics of patients from the ASLAM Healthcare Area compared to those from the rest of Galicia, there were no differences according to sex (53.3% and 53.4% were women, respectively; *p* = 0.6). On the contrary, patients attended to in the ASLAM Healthcare Area were significantly older: 46.9 (SD 24.2) vs. 46.3 (SD 23.3) years for the total number of cases; 70.4 (SD 17.6) vs. 68.8 (SD 17.43) years for hospitalized patients, and 85.4 (SD 8.8) vs. 82.9 (SD 10.2) for those who died in the hospital (*p* = 0.005, 0.006, and 0.0001, respectively).

Table 3 shows the progress of patients discharged following a first hospitalization in the ASLAM Healthcare Area compared to those in the rest of Galicia. It can be observed that patients monitored in the ASLAM Healthcare Area had a significantly shorter first hospital stay (*p* < 0.0001), a lower readmission rate (*p* = 0.05), and a shorter second hospital stay (*p* = 0.04). In the first ten months of follow-up on the discharged patients, a total of 2287 emergency department visits were recorded: 227 (9.9%) in ASLAM Healthcare Area hospitals and 2060 (90.1%) in the other centers. No differences were recorded in emergency department visits among both groups at 10, 30, and 90 days following discharge. In the first 90 days following discharge, 448 patients died due to any cause: 44 (0.39%) in the ASLAM Healthcare Area and 404 (0.43%) in the rest of Galicia (*p* = 0.2). However, analyzing only deaths due to COVID-19 in this period, it can be observed that of the 205 deaths, 9 (0.1%) were in the ASLAM Healthcare Area and 196 (2.6%) were in the rest of Galicia (*p* = 0.02).

According to our protocols, 551 of the 739 patients discharged in the ASLAM Healthcare Area should have been monitored via telemedicine. However, for reasons which remain unclear, the procedure was not followed in 76 of them and upon discharge, standard follow-up was provided. As a result, we decided to compare the progress of both groups (Table 4). Regarding baseline characteristics, the group not monitored via telemedicine was older (*p* = 0.03), but no significant differences were observed in their underlying diseases. However, there was a higher number of readmissions (*p* = 0.003) and greater mortality (*p* = 0.02) at 90 days following discharge in the group that was not monitored via telemedicine.

The TELEA protocols were appropriately complied with in 446 cases (93.9%), complied with inconsistently but in a clinically useful manner in 21 cases (4.4%), and not appropriately complied with in 8 cases (1.7%). In the latter two groups, the reason the process was not correctly followed was due to technical issues or inappropriate use of the technology in 41.4% of cases.

During the follow-up period, no deaths and no life-threatening emergencies at home were recorded. All transfers to the hospital were done using ordinary methods and no ambulance transfers were required.

## 4. Discussion

This study shows that the use of a standardized system of telemedicine with telemonitoring for patients with COVID-19 after hospital discharge is useful and safe. In these patients, we observed a significant decline in both the readmission rate and the mean length of hospital stay during first and second hospitalizations with no increases in emergency department visits or greater mortality.

The usefulness of telemedicine programs for at-home patient monitoring has previously been demonstrated in patients with various diseases [9,10,11]. In recent months, the benefits of its use in patients with COVID-19 during the pandemic has also been reported. Most of these studies have focused on the initial monitoring of patients following diagnosis [2,3,6,8,12,13]. In this situation, it has been found to be a useful, safe, easy-to-use method that helps monitor symptoms and is able to detect cases with a greater probability of hospitalization early. On the contrary, its use in follow-up and monitoring after hospital discharge, though considered appropriate [14], has not been analyzed as frequently and the studies that have been conducted have been very heterogeneous. For example, experiences have been reported in patients discharged from the emergency department [6,15], patients who were sent to medicalized hotels [7], or patients followed-up on by volunteers via telephone [16]. There are also works that evaluate the use of telemedicine in the study and management of patients with persistent symptoms and disability following hospitalization [17] or for conducting telerehabilitation [18]. Few studies have analyzed at-home clinical progress following hospital discharge with monitoring via telemedicine and telemonitoring [5,19,20] and in these works, only a limited number of patients were monitored and only one had a control group. We have reviewed other monitoring methods but their findings are not comparable [21,22,23]. Though our study has some similarities with those published in the literature (for example, use of an app, monitoring with a daily questionnaire, oxygen saturation, and/or temperature measurements), to the extent of our knowledge, it is the only work which includes all patients discharged in a specific area and which compares its results to those obtained in surrounding areas where telemedicine was not used in a consistent manner.

In regard to the program’s clinical utility, our results are very positive. When compared to patients in the rest of Galicia, patients monitored in the ASLAM Healthcare Area had a significantly shorter first hospital stay, a lower readmission rate, and shorter second hospital stay. It was also be observed that, compared to patients in our center who were not monitored via telemedicine, the readmission rate was significantly lower. These results are consistent with and expand upon the results of other works with a similar design [5]. There are undoubtedly multiple reasons underlying these findings. First, the fact that patients had direct contact with the medical team could have meant that both the physicians and the patients willingly accepted discharge from the hospital despite the persistence of symptoms which, in other circumstances, would have prolonged the hospital stay. In addition, it allowed for providing an early response to situations which, in other cases, would have entailed an emergency room visit or even readmission (such as mild or moderate complications, questions about treatment, persistent symptoms, discomfort, etc.). Lastly, if any symptoms persisted at the end of the monitoring period or long COVID was suspected, it was possible to extend the telemonitoring and refer the patient directly to a specialized unit for this specific issue.

Furthermore, this work also found a possible decrease in mortality among patients monitored via telemedicine. Data on the cause of death were obtained from the SERGAS electronic medical records for available for a very small number of patients. Therefore, we believe these data should be analyzed with caution before drawing definitive conclusions. However, we would like to highlight that in a previous article focused on the monitoring of SARS-CoV-2-positive outpatients in our area, there appears to have been a similar effect on the mortality of the group monitored via telemonitoring with a structured program versus the other patients [3]. Therefore, it can be hypothesized that comprehensive, proactive monitoring leads to earlier evaluation and treatment and thus, entails a better prognosis.

Again, this study shows the program’s safety; no unexpected deaths were recorded nor were there any life-threatening episodes at home among the patients monitored. This finding has already been described in other articles [5,6], in which no fatal events were recorded during the monitoring period.

We would like to note that monitoring via telemonitoring has been well accepted among patients. This is reflected in the degree of adherence to the program, with even better results than those described previously [16,20]. This is probably due to the proactive monitoring by the nursing department, the brief monitoring period, and the protocol’s flexibility to adapt to each patient’s technological access and skills (use of an app, contact only via telephone, etc.). In addition, satisfaction questionnaires conducted in other studies in the literature have shown very good results [5].

This study has various strengths and weaknesses. In regard to strengths, the analysis includes all patients hospitalized due to COVID-19 in a specific area in a region of nearly 3 million people during a well-defined period of time and using a single, consistent protocol. It also features a control group comprising the rest of patients hospitalized for the same reason. It also has certain limitations. First, some of the data were gathered from administrative records and as such, the precision of the data (specifically, the cause of death) can be called into question. Second, patients who required a very long hospital stay who were considered cured were often patients with a complicated hospitalization and were thus those with more severe, complicated disease. Nevertheless, the number of patients who met this exclusion criterion was small. In addition, all were in good condition upon discharge and clinical criteria were taken into account in the decision not to include them in the telemonitoring group. Lastly, this study was conducted in a single healthcare area and by a single team of professionals. Therefore, we believe it should be replicated by other teams in different populations. For all these reasons, certain aspects of this work should be interpreted with caution, especially the findings regarding mortality.

In conclusion, the use of a structured, proactive telemedicine program with telemonitoring for discharged patients with COVID-19 is a useful, safe tool that reduces the mean hospital stay and readmissions of these patients. We believe that research on the use of telemonitoring programs for discharged patients with other diseases or clinical conditions should be prioritized as they are a good alternative to habitual follow-up.

## Figures and Tables

**Figure 1 jpm-12-00609-f001:**
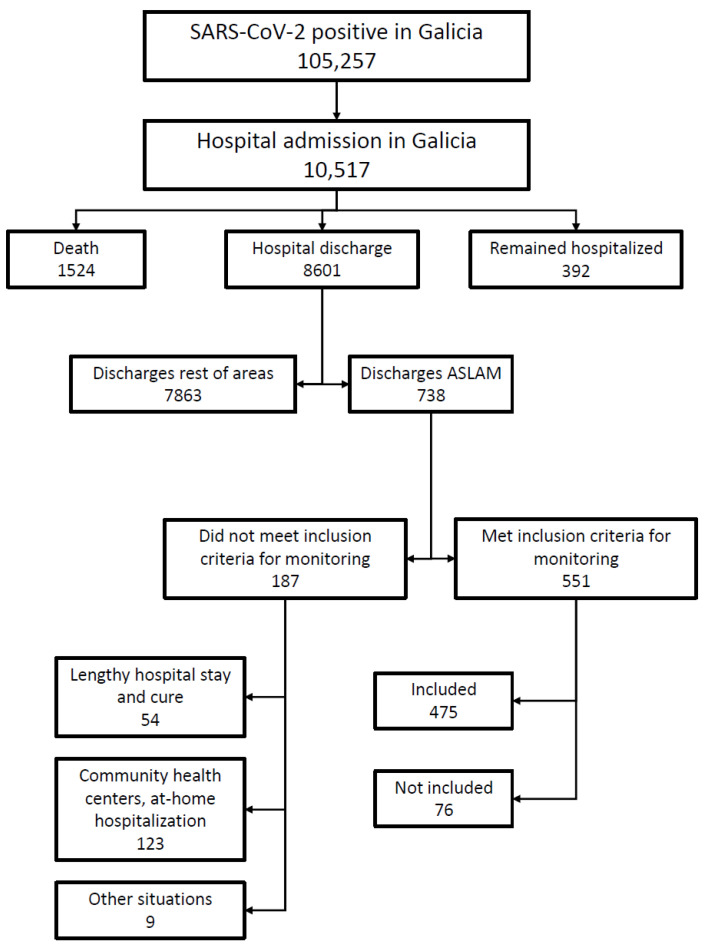
Flowchart of readmitted discharged patients in the autonomous community of Galicia.

**Table 1 jpm-12-00609-t001:** Monitoring quality criteria.

Telemedicine protocol quality criteria: Monitoring was appropriately complied with if: - At least 90% of the planned monitoring instances or contacts were conducted as scheduled. - Fewer than three consecutive monitoring instances were not conducted. - After not conducting one of the planned monitoring instances, the patient responded to a telephone call from the personnel and justified the delay. Monitoring complied with inconsistently, but clinically useful if: - At least 80% of the planned monitoring instances or contacts were conducted as scheduled. - Fewer than three consecutive monitoring instances were not conducted. - After not conducting one of the planned monitoring instances, the patient responded to a telephone call from the personnel and justified the delay. Monitoring not appropriately complied with and not clinically useful if: - Less than 80% of the planned monitoring instances or contacts were conducted as scheduled. - Three or more consecutive monitoring instances were not conducted. - After not conducting one of the planned monitoring instances, the patient did not respond to a telephone call, it was noted that he or she did not comply with the rules (for example, not remaining isolated), or he or she did not justify the delay.

**Table 2 jpm-12-00609-t002:** Characteristics of patients discharged from ASLAM hospitals.

		*n*	%
*n*		738	
Men		371	50.2%
Mean age		59.8 (SD 15.8)	
Age groups			
	<18	4	0.5%
	18–40	62	8.4%
	41–50	55	7.5%
	51–60	104	14.1%
	61–70	147	19.9%
	71–80	159	21.5%
	81–90	165	22.3%
	>90	43	5.8%
HTA		415	56.15%
Diabetes mellitus		176	23.81%
Obesity		134	18.13%
Arrhythmia		72	9.74%
Immunosuppression		52	7.03%
Non-hematologic neoplasm		45	6.08%
Ischemic heart disease		38	5.14%
COPD		38	5.14%
Heart failure		28	3.78%

**Table 3 jpm-12-00609-t003:** Evolution of discharges from ASLAM and the rest of Galicia.

		Galicia	ASLAM	*p*
		(*n* = 7863)	(*n* = 739)	
Length of previous hospital stay (days)		12.01	10.1	<0.0001
		(SD 12.14)	(SD 10.9)	
Readmission		705	50	0.05
		(8.96%)	(6.8%)	
Time until second hospitalization (days)		17.6	18.2	NS
		(SD 11.7)	(SD 9.9)	
Length of second hospitalization (days)		10.8	8.5	0.04
		(SD 10.5)	(SD 7.5)	
Mortality in second hospitalization		123	11	NS
		(17.4%)	(23%)	
Emergency department visits				
	10 days	733	82	NS
	30 days	1242	126	NS
	90 days	1633	169	
Death at 90 days		404	44	NS
		(5.2%)	(6%)	
Death due to COVID-19		196	9	0.02
		(2.5%)	(1.2%)	

**Table 4 jpm-12-00609-t004:** Characteristics of patients discharged in ASLAM.

		Telemonitoring		*p*
		YES	NO	
n		475	76	
Sex		247 (52%)	39 (51.3%)	
Mean age		66.5 (SD 16.1)	70.8 (SD 16,7)	0.03
Age groups				
	18–40	36 (7.6%)	5 (6.6%)	
	41–50	44 (9.3%)	6 (7.9%)	
	51–60	83 (17.5%)	8 (10.5%)	
	61–70	109 (22.9%)	12 (15.8%)	
	71–80	93 (19.6%)	20 (26.3%)	
	81–90	94 (19.8%)	20 (26.3%)	
	>90	16 (3.4%)	5 (6.6%)	
HT		264 (55.6%)	6 (7.9%)	
Diabetes mellitus		116 (24.4%)	21 (27.6%)	
Obesity		86 (18.1%)	15 (19.7%)	
Arrhythmia		43 (9.1%)	8 (10.5%)	
Immunosuppression		31 (6.5%)	6 (7.9%)	
Non-hematologic neoplasm		26 (5.5%)	5 (6.6%)	
Ischemic heart disease		24 (5.1%)	4 (5.3%)	
COPD		21 (4.4%)	5 (6.6%)	
Heart failure		19 (4%)	3 (3.9%)	
Death at 90 days		0	2 (2.6%)	0.001
Readmission		9 (1.9%)	5 (6.6%)	0.01

## Data Availability

The data presented in this study are available upon request from the corresponding author, and are not publicly available due to patient data protection.

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
