# Peer review of "Efficacy of Telemedicine and At-Home Telemonitoring following Hospital Discharge in Patients with COVID-19"

_jpm, 2022, doi:10.3390/jpm12040609_

Round 1

Reviewer 1 Report

This paper addressed an important topic on COVID-19. Using telemedicine technology to monitor patients can improve the efficiency. It is interesting to related readers.

However, the novelty of the method is not significant. Further comparison with existing monitoring methods should be added to the literature review.

The data analysis could be improved, can the death rate be predicted based on monitoring data? 

Figure 1 is not clear. The resolution should be improved.

Author Response

  1. However, the novelty of the method is not significant. Further comparison with existing monitoring methods should be added to the literature review.

The second paragraph of the Discussion addresses this aspect and points to different articles that analyse it in a variety of settings. On the other hand, we have not located any studies that, using other monitoring systems, could be comparable to ours. To point this out we have added a sentence at the end of paragraph 2 of the Discussion.

  1. The data analysis could be improved, can the death rate be predicted based on monitoring data?

This aspect was evaluated and the results were not meaningful. In this study the medical parameters in relation to COVID-19 were not useful in predicting death. The low number of deaths during follow-up has to be taken into account.

  1. Figure 1 is not clear. The resolution should be improved.

We have tried to improve the image quality as suggested by the reviewer.

Reviewer 2 Report

The authors provide compelling evidence on the use of telemedicine in reducing mean length of hospital stay and readmissions in COVID patients. The study is important and relevant and worthy of publication.

Author Response

We greatly appreciate your interest in our work.

Reviewer 3 Report

It is a carefully designed study which will contribute to the literature.  Some minor revisions will be helpful to improve the article further;

 1. The environment in which the study has been conducted should be described in more detail.

2. More details for statistical tests should be added. (which test used for variables)

3. Please describe the method you confirm the presence of comorbid physical disorders you listed in tables. 

Author Response

  1. The environment in which the study has been conducted should be described in more detail.

For reasons of limited space and because they are referenced in previous articles, these aspects have been omitted. In order to improve the text and its comprehensibility, the following section has been modified: "2.1 Study design and setting:".

  1. More details for statistical tests should be added. (which test used for variables)

Due to some mistakes in the transfer of information contained in the tables, certain errors had occurred, which have been corrected.

  1. Please describe the method you confirm the presence of comorbid physical disorders you listed in tables.

The method used to determine the underlying chronic pathologies was a review of the reports and discharge notes. In order to correct this lack of information pointed out by the Reviewer, we have expanded section 2.3 Inclusion and exclusion criteria.

Reviewer 4 Report

Dear authors,

It was a pleasure reading this manuscript. I hope that telemedicine will have an impact on the future and will enrich patient care. 

Author Response

(The authors gave the same response as above.)
